# Investigating the effect of dependence between conditions with Bayesian Linear Mixed Models for motif activity analysis

**Simone Lederer**[1,2]*, **Tom Heskes**[1], **Simon J. van Heeringen**[2]*, **Cornelis A. Albers**[2¤]

**1** Data Science, Radboud University, Institute for Computing and Information Sciences, Nijmegen, The Netherlands, **2** Molecular Developmental Biology, Radboud University, Research Institute for Molecular Life Sciences, Nijmegen, The Netherlands

¤ Current address: Euretos, Utrecht, The Netherlands
* slederer@cs.ru.nl (LS); s.vanheeringen@science.ru.nl (VHS)

## Abstract

### Motivation

Cellular identity and behavior is controlled by complex gene regulatory networks. Transcription factors (TFs) bind to specific DNA sequences to regulate the transcription of their target genes. On the basis of these TF motifs in cis-regulatory elements we can model the influence of TFs on gene expression. In such models of TF motif activity the data is usually modeled assuming a linear relationship between the motif activity and the gene expression level. A commonly used method to model motif influence is based on Ridge Regression. One important assumption of linear regression is the independence between samples. However, if samples are generated from the same cell line, tissue, or other biological source, this assumption may be invalid. This same assumption of independence is also applied to different yet similar experimental conditions, which may also be inappropriate. In theory, the independence assumption between samples could lead to loss in signal detection. Here we investigate whether a Bayesian model that allows for correlations results in more accurate inference of motif activities.

### Results

We extend the Ridge Regression to a Bayesian Linear Mixed Model, which allows us to model dependence between different samples. In a simulation study, we investigate the differences between the two model assumptions. We show that our Bayesian Linear Mixed Model implementation outperforms Ridge Regression in a simulation scenario where the noise, which is the signal that can not be explained by TF motifs, is uncorrelated. However, we demonstrate that there is no such gain in performance if the noise has a similar covariance structure over samples as the signal that can be explained by motifs. We give a mathematical explanation to why this is the case. Using four representative real datasets we show that at most $\sim$ €‹40% of the signal is explained by motifs using the linear model. With these data there is no advantage to using the Bayesian Linear Mixed Model, due to the similarity of the covariance structure.

**Data Availability Statement:** The project implementation is available on GitHub at https://github.com/Sim19/SimGEXPwMotifs. H3k27ac ChIP-sequencing files are available from the

BLUEPRINT project (doi:10.3324/haematol.2013.
094243) with an analysis conducted in doi:10.
1101/474403 The GTEx data is available from the
Genotype Tissue Expression database (https://
gtexportal.org/home/, project number SRP
012682) The Cacchiarelli expression data is
available from Cell (https://www.cell.com/cms/10.
1016/j.cell.2015.06.016/attachment/97ec2bc2-
5577-4d4a-966b-3cd2a63a76c2/mmc2.xlsx). The
Toufighi dataset is available from https://doi.org/
10.1371/journal.pcbi.1004256.s022 The DNase1
accessible regions used for motif calling on the
Toufighi and Cacchiarelli datasets are available
from Wouter Meuleman's website from the Altius
Institute for Biomedical Sciences, US (https://www.
meuleman.org/project/dhsindex/).

**Funding:** This work has been supported by the
Radboud University. SJvH was additionally
supported by the Netherlands Organization for
Scientific research (NWO grant 016.Vidi.189.081,
https://www.nwo.nl/en). CAA was supported by the
FP7-PEOPLE-2013-CIG program (project nr.
631716).

**Competing interests:** The authors declare no
conflict of interest.

## Availability & implementation

The project implementation is available at https://github.com/Sim19/SimGEXPwMotifs.

## Introduction

Cell type-specific gene expression programs are mainly driven by differential expression and binding of transcription factors (TFs). The human genome contains $\sim 1,600$ TFs, which represent 8% of all genes [1]. These proteins bind DNA in a sequence-specific manner and typically have a 1000-fold or greater preference for their cognate binding site as compared to other sequences [2]. By binding to cis-regulatory regions, i.e. promoters and enhancers, they can control the chromatin environment and the expression of downstream target genes [1]. Cell type identity is determined by the expression of a select number of TFs. This is evidenced by the growing number of cell reprogramming protocols that rely on the activation of a few TFs to reprogram the cell state, for instance from a somatic cell to a pluripotent stem cell [3, 4]. Mis-regulation of TF expression or binding is associated with a variety of diseases, such as developmental disorders and cancer [3]. Hence, it is of great importance to understand the mechanisms of gene regulation driven by TFs.

TFs bind to specific DNA sequences called sequence motifs. These motifs are relatively short, with a length usually ranging from six to twelve nucleotides, and flexible in the sense that several TFs can bind to the same motif [1]. The binding sites of TFs can be determined genome-wide using chromatin immunoprecipitation with specific antibodies followed by high-throughput sequencing (ChIP-seq). Although ChIP-seq studies suggest that many TF binding events appear to be not functional, the presence of a sequence motif is still predictive of gene expression [1]. With a linear regression model, in which the sequence information is used to model gene expression, one can learn the TFs that play a major role in gene regulation [5–9]. Typical approaches either use linear regression with $\mathcal{L}^2$-regularization (Ridge Regression) or a combination of $\mathcal{L}^1$- and $\mathcal{L}^2$-regularization (ElasticNet). These approaches tend to explain only a small fraction of the variation of gene expression. However, due to the large number of genes, the coefficients are generally highly significant and can be interpreted as a measure of TF activity.

One of the key assumptions of a linear regression model is the independence between samples. If samples originate from the same cell line, tissue, or other biological source, this assumption may be invalid. In addition, related cell types will also have similar gene expression profiles, where the expression of many genes will be highly correlated.

Here, we propose a Bayesian Linear Mixed Model that builds upon the previously described Bayesian Ridge Regression [5, 6], but allows for correlated motif activity between samples. Our model relaxes the rigid independence assumption common to earlier approaches Fig 1.

We compare our full Bayesian Linear Mixed Model with Bayesian Ridge Regression on simulated data, for which we control the degree of correlation between samples. We show that the Bayesian Linear Mixed Model formulation outperforms the Ridge Regression for data with randomly distributed noise. This is the case especially for highly correlated data. We further show that the Bayesian Linear Mixed Model loses its superiority over the Ridge Regression if only a small part of the gene expression signal can be explained by motif influence, while other influential factors contribute largely to the gene expression. We confirm the observations made during the simulation study on four real-world datasets, in which a significant amount of the biological signal cannot be uniquely explained by a linear combination of motifs. We

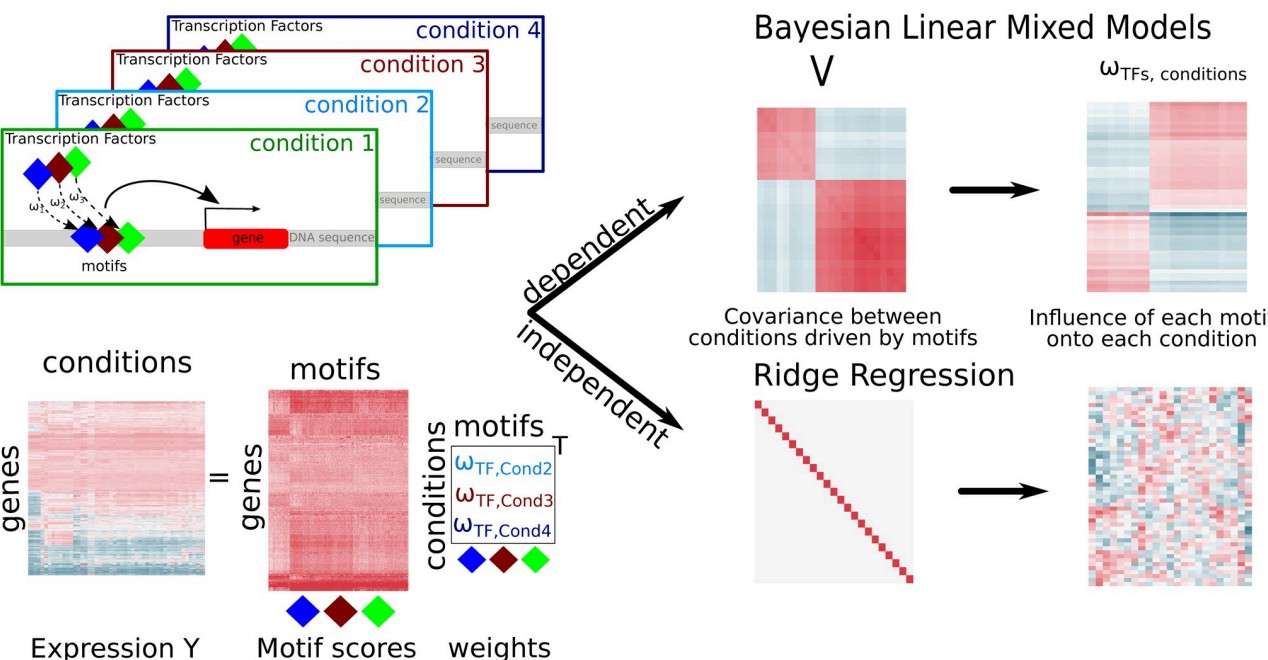

**Fig 1. Project overview.** We model the motif influence on gene expression signal in different conditions with a linear combination of the motif scores in each promoter region of the gene. We introduce the Bayesian Linear Mixed Model that allows for correlation between samples in contrast to the Ridge Regression, which is commonly used to model motif activity.

can explain this phenomenon mathematically and give more technical details regarding the computation of the model.

## Methods

Here, we introduce the mathematical models that are used throughout this paper. First, we introduce the linear model that represents the signal of expression data $\mathbf{Y}_{G,C}$ as a linear combination of motif scores and their influential weights. Second, we present a Bayesian perspective on the model. Finally, we show that the Ridge Regression formulation is a special case of the newly introduced Bayesian formulation. We then outline in detail how we simulate data, which we use to compare the complete Bayesian perspective and the Ridge Regression model.

In the following sections we will interchangeably use sample or condition.

### Inference of motif activities

The general model used throughout this paper models gene expression $\mathbf{y}_{g,c}$ in condition $c$ as a linear function of motif scores $\mathbf{m}_{t,g}$ of motifs $t = \{1, \ldots, T\}$, weighted by the motif influence $\omega_{t,c}$, at the promoter region of gene $g$:

$$\bar{\mathbf{y}}_{g,c} = \sum_{t=1}^{T} \mathbf{m}_{t,g}\omega_{t,c} + \text{noise}, \tag{1}$$

with the normalized gene expression data $\bar{\mathbf{y}}_{g,c} = \mathbf{y}_{g,c} - \bar{\mathbf{y}}_g - \bar{\mathbf{y}}_c$, where $\bar{\mathbf{y}}_g$ is the average signal over the promoter of gene $g$ over all conditions $C$ and $\bar{\mathbf{y}}_c$ is the average signal of condition $c$ over all genes $G$. In the following, we will refer to the normalized gene expression simply as $\mathbf{Y}_{G,C}$. The term "noise" represents all signal that cannot be explained by the model, i.e. the

linear combination of the motif scores $\mathbf{M}_{T,G}$. This can be any technical noise, motif influence, for which the linear assumption might be too simplistic, but also any other source that drives the gene expression $\mathbf{y}_{g,c}$ and is not modeled. The model was originally introduced by [5] and subsequently expanded by [6].

For the application of the above model to the H3K27ac dataset, we re-use the above notation, where $G$ represents the dimensionality in enhancer space (not gene space). Instead of computing motif scores $\mathbf{M}_{T,G}$ in the promoter region, we compute them in the enhancer region and the expression signal $\mathbf{Y}_{G,C}$ is the H3K27ac ChIP-seq signal.

## Bayesian Linear Mixed Models

The main idea behind a Bayesian formulation is to include prior knowledge about the data, called the prior. Here, we model $\boldsymbol{\omega}_{t,c}$, the influence of motif $t \in \{1, \ldots, T\}$ in condition $c \in \{1, \ldots, C\}$, as a normally distributed prior with mean zero, with $\sigma^2 \mathbf{I}_T$ being the covariance over all motifs and $\mathbf{V}_C$ the covariance over all conditions:

$$\mathrm{vec}(\omega_{T,C}) \sim \mathcal{N}(\mathbf{0}_{TC}, \sigma^2 \mathbf{V}_C \otimes \mathbf{I}_T). \tag{2}$$

We use the vector notation for the matrix normal distribution, for which we make use of the Kronecker product between the covariance matrices. Another mathematical notation of the model is $\mathcal{MN}_{T,C}(\mathbf{0}_{T,C}, \sigma^2 \mathbf{I}_T, \mathbf{V}_C)$. Note that in the vectorized notation the mean $\mathbf{0}_{T,C}$ is written in vector notation, too: $\mathrm{vec}(\mathbf{0}_{T,C}) = \mathbf{0}_{TC}$.

In this paper, we assume independence between motifs. Extending the assumption to dependence between motifs with covariance matrix $\boldsymbol{\Psi}$ could easily be implemented.

Combining the prior knowledge about $\boldsymbol{\omega}_{T,C}$ in Eq (2) with the model in Eq (1), it follows that the expression data $\mathbf{Y}_{G,C}$ conditioned upon $\boldsymbol{\omega}_{T,C}$ obeys:

$$\mathrm{vec}(\mathbf{Y}_{G,C}|\omega_{T,C}) \sim \mathcal{N}(\mathbf{0}, \sigma^2 \mathbf{V}_C \otimes \Pi_G + \delta \Sigma_C \otimes \mathbf{I}_G), \tag{3}$$

with $\Pi_G = \mathbf{M}_{T,G}^\mathsf{T} \mathbf{M}_{T,G}$, where $\mathsf{T}$ is the transpose. Hence, the covariance between genes is driven by the similarity among motif counts $\mathbf{M}_{T,G}$. $\Sigma_C$ is the covariance of noise between conditions and $\delta \mathbf{I}_G$ is the covariance matrix between genes. Thus, we assume independence between genes in the noise term. The posterior mean of the motif influence $\boldsymbol{\omega}_{T,C}$, denoted $\hat{\omega}_{T,C}$, given the expression $\mathrm{vec}(\mathbf{Y}_{G,C})$ then reads:

$$\mathrm{vec}(\hat{\omega}_{T,C}|\mathbf{Y}_{G,C}) = (\mathbf{V}_C \otimes \mathbf{M}_{T,G}^\mathsf{T} \mathbf{I}_T)[\sigma^2 \mathbf{V}_C \otimes \Pi_G + \delta \Sigma_C \otimes \mathbf{I}_G]^{-1} \mathrm{vec}(\mathbf{Y}_{G,C}). \tag{4}$$

This model is explained in more detail in S1A Appendix.

**Special case: Ridge regression.** Ridge Regression, also known as Tikhonov Regularization, prevents over-fitting in a linear regression by using an $\mathcal{L}^2$-regularization on the estimated parameters. For more details on $\mathcal{L}^2$-regularization and Ridge Regression, we refer the reader to [10]. Note that in a Ridge Regression the samples are assumed to be isotropic, i.e. independent and identically distributed. The covariances in the model, $\mathbf{V}_C$ and $\Sigma_C$, therefore reduce to identity matrices, which only differ in the constant with which they are multiplied:

$$\mathbf{V}_C = \sigma^2 \mathbf{I}_C \quad , \quad \boldsymbol{\Sigma}_C = \delta \mathbf{I}_C. \tag{5}$$

With the above model we formulated a Bayesian Linear Mixed Model to explain the motif influence on expression data $\mathbf{Y}_{G,C}$ based on motif scores. The Bayesian Linear Mixed Model provides a relaxation of the so far used assumption of independence in Ridge Regression. It allows for modeling dependency structures between conditions and in the noise, which can increase the power of the model, as shown in the Results section. In the following, we will refer

to the model as Bayesian Linear Mixed Model when allowing for correlation between samples and noise, i.e. the covariance matrices $\mathbf{V}_C$ and $\boldsymbol{\Sigma}_C$ are not restricted except for being symmetric positive definite matrices. We refer to the model as Ridge Regression when we specifically assume independence between conditions and noise, i.e. $\mathbf{V}_C = \boldsymbol{\sigma}^2 \, \mathbf{I}_C$, and $\boldsymbol{\Sigma}_C = \boldsymbol{\delta} \, \mathbf{I}_C$.

## Code availability

The project implementation (https://github.com/Sim19/SimGEXPwMotifs) is provided in the programming language Python [11, 12], where we make use of the packages `pandas` [13], `matplotlib` [14], `seaborn` [15] and `numpy` [16].

## Model fitting

We run the optimization of Eq (3) to compute Eq (4) with the Python package `limix` [17–19] and use the module `VarianceDecomposition()`, which we use with the parameter settings for restrictions on $\mathbf{V}_C$ to "freeform", which models concurrently $\boldsymbol{\Sigma}_C$ to be of "random" shape. These parameter settings only restrict $\mathbf{V}_C$ and $\boldsymbol{\Sigma}_C$ to be positive-definite matrices, with $\frac{1}{2} C \times (C + 1)$ parameters to be estimated. With these settings, there are no restrictions on the rank of the matrix, i.e. the degree of correlation among samples. For the optimization start, $\mathbf{V}_C$ and $\boldsymbol{\Sigma}_C$ are both set to be the estimated covariance matrix of $\mathbf{Y}_{G,C}^{\top}$, each divided by half. The optimization is based on the *L-BFGS* algorithm to minimize the log likelihood with an $\mathcal{L}^2$-regularization along the non-diagonal elements of the covariance matrix $\boldsymbol{\Sigma}_C$, which is also known as isotropic Gaussian prior with zero mean. The implementation makes use of the reduction of computational complexity by using the Kronecker product notation and its identities for the case of matrix variate data, which is highly efficient [18, 19]. For more detailed information about Linear Mixed Models and its implementation in `limix`, refer to [20]. We run Ridge Regression in the simulation study with the Python implementation in the package `sklearn.linear_model.RidgeCV()` [21]. For the data application, we make use of the `limix.VarianceDecomposition()` implementation with the setting "freeform" for $\mathbf{V}_C$ and "random" noise for $\boldsymbol{\Sigma}_C$ for fitting the Bayesian Linear Mixed Model. For the computation of Ridge Regression, we restrict $\mathbf{V}_C$ and $\boldsymbol{\Sigma}_C$ to be identity matrices. Both computations (`limix.VarianceDecomposition()` with restrictions for $\mathbf{V}_C$ and $\boldsymbol{\Sigma}_C$ to identity matrix and `sklearn.linear_model.RidgeCV()`) yield the same results (see S1 Fig).

## Visualization

For the visualizations in this article, we work with the plotting facility from `pandas` and `seaborn` [15]. For boxplots and other results from the simulation study, we make use of the `R-ggplot2` package [22] and `R-cowplot` [23]. The visualization of clustered data is done with python's `seaborn.clustermap()` using the "complete" method for the dendogram computation on Euclidean distance.

## Simulating data

For the simulation study we generate data based on the model introduced in Eqs (1)–(4), given a covariance matrix $\mathbf{V}_C$. The prior weight $\tilde{\omega}_{T,C}$ and the expression signal $\mathbf{Y}_{G,C}$ are generated according to Eqs (2) and (1). The expression data $\mathbf{Y}_{G,C}$ from Eq (1) is then used to estimate $\mathbf{V}_C$ and $\boldsymbol{\Sigma}_C$. Based on these computations, the posterior motif influence $\hat{\omega}_{T,C}$ is then computed and compared to the simulated motif influence $\tilde{\omega}_{T,C}$ with a Pearson correlation [24] over all conditions. We provide the same analysis with the Spearman's rank correlation [25] and the Mean-Squared Error S1 and S3 Figs. As gene set we use the 978 landmark genes from the LINCS

project [26]. In a secondary simulation we increase the size of the gene set to 5000 genes, which originate from an analysis of the most variational genes across all samples from the GTEx project (Genotype Tissue Expression, https://gtexportal.org/home/). We generate data for $C = \{10, 30, 50, 70, 100, 120\}$ conditions.

## Covariance types of $\mathbf{V}_C$

For the generation of simulated motif influence $\tilde{\omega}_{T,C}$ (Eq (2)), we need to give a covariance matrix $\sigma^2 \mathbf{V}_C \otimes \mathbf{I}_T$. As the covariance along TFs is modeled to be isotropic (independent and identically distributed), one can generate $\tilde{\omega}_{t,c}$ randomly $T$ times with $\tilde{\omega}_{t,c} \sim \mathcal{N}(0, \sigma^2 \mathbf{V}_C)$. For the covariance between the weights $\tilde{\omega}_{T,C}$ in Eq (2) and hence expression data $\mathbf{Y}_{G,C}$ in Eq (3), we consider four types of covariance matrices for $\mathbf{V}_C$, according to assumptions made about the correlation between samples: (i) Isotropic V: same number on the diagonal, all off-diagonal elements set to zero. Samples drawn from such a covariance matrix are independent. This matches the implicit assumption of the Ridge Regression. We will refer to this as "V: independent". (ii) Full matrix V with all positive off-diagonal elements. Samples drawn from such a covariance matrix are positively correlated, without any specific (block) structure. This is a clear mismatch with the assumptions underlying Ridge Regression, and results are expected to favor the (full) Bayesian Linear Mixed Model. We will refer to this as "$\mathbf{V}_C$: correlated (no groups)". (iii) Block structure with many blocks, each modeling a group of biological replicates. Samples within each block are highly correlated, with small correlation values between the blocks. Blocks vary in size. Average block size is two. We will refer to this as "$\mathbf{V}_C$: correlated (many groups)". (iv) Block structure similar to (iii), but now with just two blocks. We will refer to this as "$\mathbf{V}_C$: correlated (two groups)".

For the generation of the correlated covariance matrices with groups, we provide pseudocode in S2B Appendix. An exemplary visualization of a covariance matrix with samples that correlate in many groups is given in Fig 2A and 2B, left panel.

## Noise $\Sigma_C$–unstructured and structured

To generate the gene expression data $\mathbf{Y}_{G,C}|\tilde{\omega}_{T,C}$, we compute the signal as the product of the motif scores $\mathbf{M}_{T,G}$ (explained hereafter) and their weights $\tilde{\omega}_{T,C}$ (explained previously). Due to the randomness in the signal that is not explained by motifs, we add some random noise, which is drawn from a normal distribution with covariance $\delta \Sigma_C \otimes \mathbf{I}_G$. We generate $\Sigma_C$ in two different ways: (i) we assume no particular structure, $\Sigma_{C,\mathrm{random}}$, which is a matrix filled with values drawn from a standard normal distribution, multiplied with itself. This is the Wishart distribution with $C$ degrees of freedom. (ii) We add the same covariance matrix $\mathbf{V}_C$ that is used to model the correlations between the conditions to $\Sigma_{C,\mathrm{random}}$ from (i): $\Sigma_{C,\mathbf{V}_C}, \rho = \Sigma_{C,\mathrm{random}} + \eta_\rho \mathbf{V}_C$. The noise matrices are then normalized by their trace. The detailed description of the generation of the noise matrices as well as the control of structuredness in it is given in S2B Appendix. For the simulation study depicted in Fig 2 and discussed in the Results section we generate the structured noise with $\rho = 0.7$. A visualization of both types of noise matrices is given in Fig 2A and 2B, both in the respective right panel.

**Signal-to-noise ratio.** Previous research has shown that roughly 10-20% of the signal of gene expression can be explained by motif influence in the promoter region [6]. We therefore generate the data in such a way, that 20% of the signal in expression data $\mathbf{Y}_{G,C}$ is due to motifs, and the rest unexplainable noise. We achieve this by adjusting the parameter $\sigma^2$ and $\delta$. We fix $\delta$ and determine $\sigma^2$ by bisection such that it explains 0.2 of the variance coefficient (Eq (1)). More details can be found in S2B Appendix.

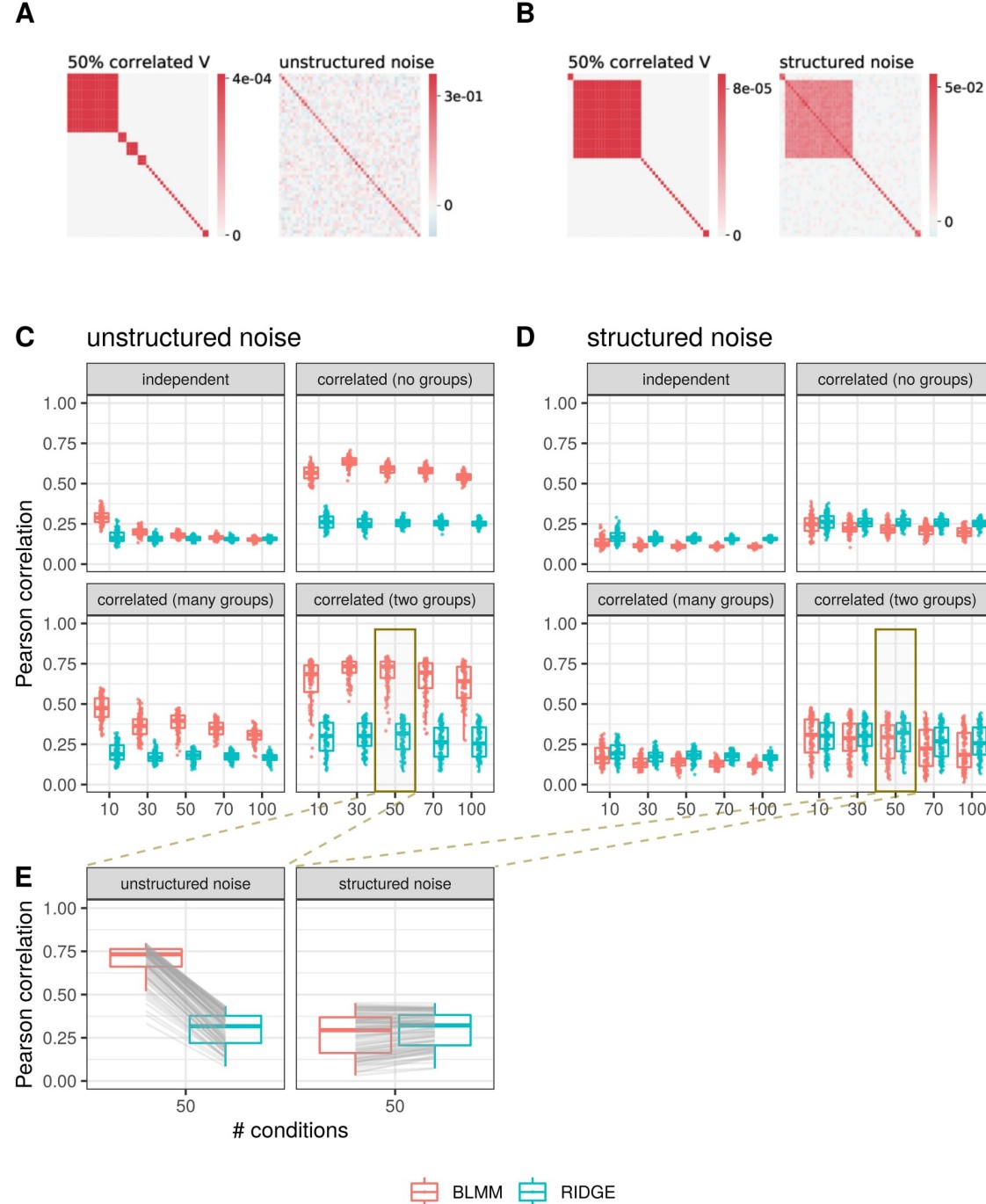

**Fig 2. Simulation study.** Data is generated over $G = 978$ informative genes and $T = 623$ motif scores and 100 repetitions, with either unstructured noise $\Sigma_C$ (A,C) or structured noise $\Sigma_C$ (B,D). (A,B): exemplary covariance matrix between samples, $\mathbf{V}_C$, and noise matrix $\Sigma_C$ being unstructured (A) or structured (B). Data is shown for $C = 50$ samples with a covariance between conditions with $k = 25$ blocks, i.e. 25 sample groups, in which samples are completely correlated. (C) and (D): The Pearson correlation values between generated and estimated motif-condition-weights are shown for different degrees of correlation between samples. The values are depicted per method used to predict the motif-condition-weights: the Bayesian Linear Mixed Model (BLMM, in red), which allows for dependence between samples, and the Ridge Regression (RIDGE, in blue), which assumes independence between samples. Data is generated with the following covariances between samples. (i) independence between samples, $\mathbf{V}_C = \mathbf{I}_C$, (ii) unrestricted correlation between samples, (iii) correlated data with many sample groups, and (iv) highly correlated samples, assuming samples originate from two biologically different samples. Results are shown based on data generated with unstructured noise ((C), see (A)), or with structured noise ((D), see (B)). (E): Exemplary comparison of Pearson correlation values between simulated and predicted posterior motif influence $\hat{\omega}_{T,C}$. The data is generated with highly correlated samples, modeling two groups of biological replicates. Corresponding replicates are combined by a gray line. The data in the left panel is generated with unstructured noise $\Sigma_C$ (see (A)) and in the right panel with structured noise (see (B)).

## Motif scores

For the computation of the TF motif scores $\mathbf{M}_{T,G}$ we use log-odds scores based on the positional frequency matrices, which are computed with the software `GimmeMotifs`, v13 [27, 28]. We make use of the database `gimme.vertebrate.v3.1`, included with `GimmeMotifs`. In general, we filter for genes that are known to be protein coding and on chromosomes 1-22 and X. For the GTEx dataset we assume the promoter region to be 400bp upstream and 100bp downstream of the Transcription Start Site of the given gene sets. We take the Transcription Start Site from the GENCODE database (version 26) [29] and generate a `bed` file with the promoter regions with `BEDTools, v2.17.0` [30] and the subcommand `slop`. For the Cacchiarelli and Toufighi dataset, we scan for motifs in the closest accessible region, as measured using DNase1. We downloaded the regulatory index (version 20190703), created by integrating 733 DNase-seq datasets [31], from https://www.meuleman.org/project/dhsindex/. For each gene we select the DNaseI summit with the highest mean signal within 1kb of the gene Transcription Start Site. We use 200bp centered at the summit for motif scanning. For the H3K27ac data we use 200bp centered at the summit of the corresponding DNaseI peak, see [28] for more details.

## Cross-validation and permutation

In the Results section, we compare the performance of the Bayesian Linear Mixed Model and of the Ridge Regression on four real-world datasets. To assess the performance of both model assumptions, we make use of a ten-fold cross-validation by creating ten random subsets across the genes. Each of these ten subsets is used as a test dataset, while the model is trained on the union of the remaining nine subsets. As there is no knowledge about the motif influence $\tilde{\omega}_{T,C}$, we compute the expression $\hat{\mathbf{Y}}_{G,C}$ with the predicted posterior motif influence $\hat{\omega}_{T,C}$ (Eq (1)) and compare the Pearson correlation between predicted expression $\hat{\mathbf{Y}}_{G[\text{test}],C}$ on the test set and the original expression $\mathbf{Y}_{G[\text{test}],C}$ of the test set.

Per cross-validation round, we additionally run 1000 permutations (without replacement) in the motif scores $\mathbf{M}_{T,G}$ along the genes.

## Experimental data

For the application analysis presented in the Results section, we make use of four experimental datasets.

**H3K27ac ChIP-sequencing signal at hematopoietic enhancers.** We use experimental enhancer activity data from the human hematopoietic lineage [28, 32]. This dataset is based on 193 ChIP-seq experiments in 33 hematopoietic cell types using an antibody specific for histone H3 acetylated at Lysine 27 (H3K27ac). The H3K27ac histone modification is deposited by the histone acetyltransferase p300 (EP300) and is associated with enhancer activity [33]. ChIP-seq reads were counted in 2kb regions centered at accessible regions, log-transformed and normalized using scaling by Z-score transformation. The peaks, or accessible regions, represent putative enhancers. We subset the peaks to the most variable 1000 peaks over all samples. We selected all replicates of the cell types "monocytes" and "T-cells", which are 23 samples in total. Different samples of the same cell type represent different donors.

**Human tissue gene expression data (GTEx).** Second, we make use of gene expression data from human tissues. The data is from the Genotype Tissue Expression (GTEx) database [34], and is available on https://gtexportal.org/home/. It is RNA-seq data from many different tissues. The data was downloaded with project number *SRP*012682 with the R-package `R-recount, v.1.63` [34–37]. We scale the raw counts by the total coverage of the sample

(function `scale_counts()`, setting "by = auc") and keep entries with at least 5 counts. We transform the data with the `DESeq2`-package, `v.1.20.0` and use the variance stabilizing transformations [38–40], implemented in the package function `vst()`, with `blind` transformations to the sample information.

We selected the 5, 000 most variable genes over all samples of the entire GTEx experiment. We then choose 75 random samples, of which there are 35 different tissues from 21 different organs.

As we model the normalized expression data $\mathbf{Y}_{G,C}$ (see Eq (1)), we subtract the mean along genes and along conditions from the expression data. For the analysis, we further normalize the motif score matrix.

**Time-series data.**  We additionally compare the performances of both model assumptions on two time-series datasets [41, 42]. Samples from different time-series are known to be highly correlated.

**Human cellular reprogramming RNA-seq data.**  We make use of RNA-seq data for a reprogramming time-course from human induced fibroblasts (hiF) to induced pluripotent stem cells (hIPSC) [41]. Gene expression levels were measured at several different time points, and human embryonic stem cells (hESC) were included for comparison. The expression data is available at https://www.cell.com/cms/10.1016/j.cell.2015.06.016/attachment/97ec2bc2-5577-4d4a-966b-3cd2a63a76c2/mmc2.xlsx. We transform the data to a log2-scale. For more details on the data and the original analysis, refer to [41].

**Human keratinocyte differentiation microarray data.**  The microarray data of the differentiation of human primary keratinocytes was originally generated by [43] and is re-used in [42]. Gene expression levels were measured every five hours over a time span of 45h, resulting in ten samples. Measurements were taken in triplicates. Data is available at the article's website and published at https://doi.org/10.1371/journal.pcbi.1004256.s022. The data has been processed using background correction and quantile normalization, and was log2-transformed [42].

## Results

In this section, we apply the model that we introduced in detail in Eq (1)—Eq (4), to simulated data and to four real-world datasets. We compare the two assumptions about the shape of covariance, as discussed before: the novel allowance of dependence (Bayesian Linear Mixed Model) to the restriction of independence, that has been applied so far (Ridge Regression).

### Simulation

To quantify the differences in the model when allowing for dependence between conditions, instead of assuming independence, we simulate data according to Eq (1). We generate different datasets for $G = 978$ genes, for $C = \{10, 30, 50, 70, 100, 120\}$ samples, and $T = 623$ motifs. We vary the degree of correlation between the samples, expressed in the covariance matrices $\mathbf{V}_C$ and $\Sigma_C$. We also vary the influence of these covariance matrices on the signal: As covariance matrix $\mathbf{V}_C$ we generate (i) an identity matrix, (ii) a full matrix with positive off-diagonal elements (unrestricted correlation between samples), (iii) a block matrix with $k = \frac{1}{2}C$ blocks along the diagonal, which models groups of biological replicates and (iv) a block-structured matrix matrix with $k = 2$ blocks along the diagonal, modeling $k = 2$ groups of biological replicates. As noise matrix $\Sigma_C$ we first generate unstructured or random noise. In a second step, we generate the data with a structured noise matrix $\Sigma_C$. For every parameter set we generate 100 replicates to verify the robustness of the two model assumptions. We compare the performance of the two assumptions on the shape of covariance with a Pearson correlation score.

The correlation is computed between the simulated motif influence $\tilde{\omega}_{T,C}$ and the estimated posterior motif influence $\hat{\omega}_{T,C}$. The higher the correlation values the better the performance of the model. In S1 Fig, we confirm that the Bayesian Linear Mixed Model with fitting independent samples and independent noise is equal to Ridge Regression.

We summarize the simulation study visually in Fig 2.

We separate the simulation study first on data generated with unstructured noise (exemplary visualization of data in Fig 2A, Pearson correlation values in Fig 2C) and second on data with structured noise (exemplary data shown in Fig 2B, model performances shown in Fig 2D). Summarizing, we compare the performance of both models on the two types of data (unstructured versus structured noise $\Sigma_C$) in Fig 2E. We show an example of the model performance for data generated with a covariance matrix $\mathbf{V}_C$ that models all samples to originate from many blocks, on unstructured (left) and structured noise (right). In S5 Fig, we show on the example of $C = 50$ samples that a higher dimensional dataset, generated with $G = 5000$ genes, exhibits the same performance as for $G = 978$ genes.

**Unstructured noise $\Sigma_C$.** Given the covariance $\mathbf{V}_C$, we generate a gene expression signal $\mathbf{Y}_{G,C}$ according to Eq (1) with random noise $\Sigma_{C,\text{random}}$ (see Fig 2A). In Fig 2C, we compare the performance of the Bayesian Linear Mixed Model, which allows for dependencies between samples, to Ridge Regression, which assumes independence. We depict the Pearson correlation values between estimated and simulated motif influence $\boldsymbol{\omega}_{T,C}$ for 100 randomly generated datasets per boxplot. In every panel, we compare both model assumptions, (i) dependence (in red, labeled BLMM) and (ii) independence (in blue, labeled RIDGE). We present the results separated by the degree of correlation in $\mathbf{V}_C$, which was used to generate the data. Allowing for dependencies between conditions leads to a better prediction performance, especially for correlated data (Fig 2C and 3E), independent of the sample size $C$. For uncorrelated data ($\mathbf{V}_C = \sigma^2 \mathbf{I}_C$), the allowance for dependency yields a slightly better performance for small sample sizes, $C \in \{10, 30, 50\}$ (Fig 2C, upper left panel). For higher sample sizes, the performances are equal. In Fig 2E, left panel, we show explicitly that the Pearson correlation values that result from the dependence assumption (BLMM, in red) are always higher than those from the independence assumption (RIDGE, in blue). The exemplary visualization shows the correlation between estimated and simulated motif influence $\tilde{\omega}_{T,C}$ that was generated with a covariance matrix $\mathbf{V}_C$ with two blocks along the diagonal, i.e. $k = 2$ correlated groups of samples over $C = 50$ conditions. We run both model assumptions on the same datasets. The correlation values from the same datasets, depicted per model assumption, are connected with gray lines.

**Structured noise $\Sigma_C$.** For the generation of the gene expression signal with structured noise, we add the structure of correlation between conditions to the noise (see Fig 2B). This is motivated by the fact that we can explain roughly 20% of the signal in expression data $\mathbf{Y}_{G,C}$ data by motif influence, but not the remaining 80% of the signal. This remaining signal is often similar to the covariance between samples. In Fig 2D, we depict the Pearson correlation values that are computed between simulated motif influence $\tilde{\omega}_{T,C}$ and posterior motif influence $\hat{\omega}_{T,C}$. They result from applying the dependence (BLMM, in red) and independence assumption (RIDGE, in blue) to data that is generated with such a structured noise at a degree of $\boldsymbol{\rho} = 0.7$. The results are again shown for differently correlated datasets between conditions $\mathbf{V}_C$, analogous to the previous section. With that structure in the noise $\Sigma_C$, both methods perform equally with low correlation values. As shown in Fig 2E, right panel, the Bayesian Linear Mixed Model performs equally or slightly worse than Ridge Regression.

Indeed, when applying a degree of structure in the noise $\Sigma_C$ (explained in detail in Eq. (17) and Eq. (18) in S2B Appendix) by varying $\boldsymbol{\rho}$ between zero and one in a step size of 0.1, there is loss of performance of the Bayesian Linear Mixed Model for an increasing structured signal in

the noise (see S4 Fig). In contrast, the correlation values resulting from Ridge Regression applied to this data with structured noise are very similar, if not equal, to those from data that was generated with unstructured noise (Fig 2C). Both model assumptions perform approximately equally when applied to data with structured noise. For an exemplary visualization we depict the correlation values per sample in Fig 2E, right panel, with lines connecting the two results from the two model assumptions, which were applied to the same dataset. The reason for this performance loss is the computation of the posterior motif influence $\hat{\omega}_{T,C}$ (Eq (4)). If $\Sigma_C$ takes a structure that is too similar to $\mathbf{V}_C$, these two terms can be summarized in the covariance of vec($\mathbf{Y}_{G,C}|\boldsymbol{\omega}_{T,C}$) in Eq (3). This covariance returns as inverse in Eq (4) and therefore $\mathbf{V}_C$ and $\Sigma_C$ cancel out with $\mathbf{V}_C$ from the motif-dependent term $\mathbf{V}_C \otimes \mathbf{M}_{T,G}^{\mathsf{T}}\mathbf{I}_T$. We give more mathematical details in the Discussion.

## Application

We compare the Bayesian Linear Mixed Model and the Ridge Regression on four real-world datasets. First, we apply the method to a ChIP-seq dataset of the histone modification H3K27ac in the human hematopoietic lineage. This histone mark correlates with enhancer activity [33]. We refer to this dataset as H3K27ac. Second, we use RNA-seq gene expression data from the GTEx consortium. It is known from RNA-seq data, that there is generally a high contribution of "technical" noise, such as measurement noise that is introduced purely by the experiment (laboratory [44, 45] and batch [46]) and by biological variation [47]. Hence, we expect a significant contribution of such "technical" noise to the signal. We refer to this RNA-seq dataset as GTEx. We additionally compare the two model assumptions on two time-series datasets: the Cacchiarelli dataset is an RNA-seq dataset applied to reprogramming from fibroblasts to IPSCs. Additionally, we analyse microarray data, which we refer to as Toufighi data, that measures the human keratinocyte differentiation over 45 hours. For all four datasets, we compare the dependence and independence assumption (Bayesian Linear Mixed Model and Ridge Regression) by means of a cross-validation.

**Acetylation data H3K27ac.** We split the acetylation dataset H3K27ac into two sample groups, originating from two different cell types: (i) monocytes from the myeloid lineage and (ii) T-cells from the lymphoid lineage. Within these two groups the ChIP-seq signal $\mathbf{Y}_{G,C}$ is highly correlated as the samples represent the same cell types from different donors. The results from the application of the Bayesian Linear Mixed Model and the Ridge Regression to the H3K27ac dataset are summarized in Fig 3.

To assess and compare the model performances, we run a ten-fold cross-validation. Per run we compute the Pearson correlation between measured and predicted ChIP-seq signal $\mathbf{Y}_{G,C}$ of the test and training set. The correlation values from the cross-validation are shown in Fig 3B. The performance of the test set (solid line) is depicted to gain insight into the prediction performance of each model, the performance of the training set (dotted line) to check for over-fitting. We additionally depict a summary of Pearson correlations over 1000 permutations in the motif scores $\mathbf{M}_{T,G}$ along the genes $G$ per cross-validation (gray line). Both model assumptions yield very similar performances and shuffling the motif scores completely negates the models' performances. Hence, on the basis of cross-validation, no clear statement about superiority of one model assumption to another can be made. A visualization of the estimated $\mathbf{V}_C$ and $\Sigma_C$, depicted in S6–S9 Figs, emphasizes again the great difference between the two models and their model fits. While for Ridge Regression both matrices are identity matrices with a scaling factor, both covariance matrices from the more flexible Bayesian Linear Mixed Model exhibit strong correlation structures (S8 and S9 Figs). These two latter matrices are very similar. Clustering the covariance matrices yields two clearly separated blocks along the diagonal, showing

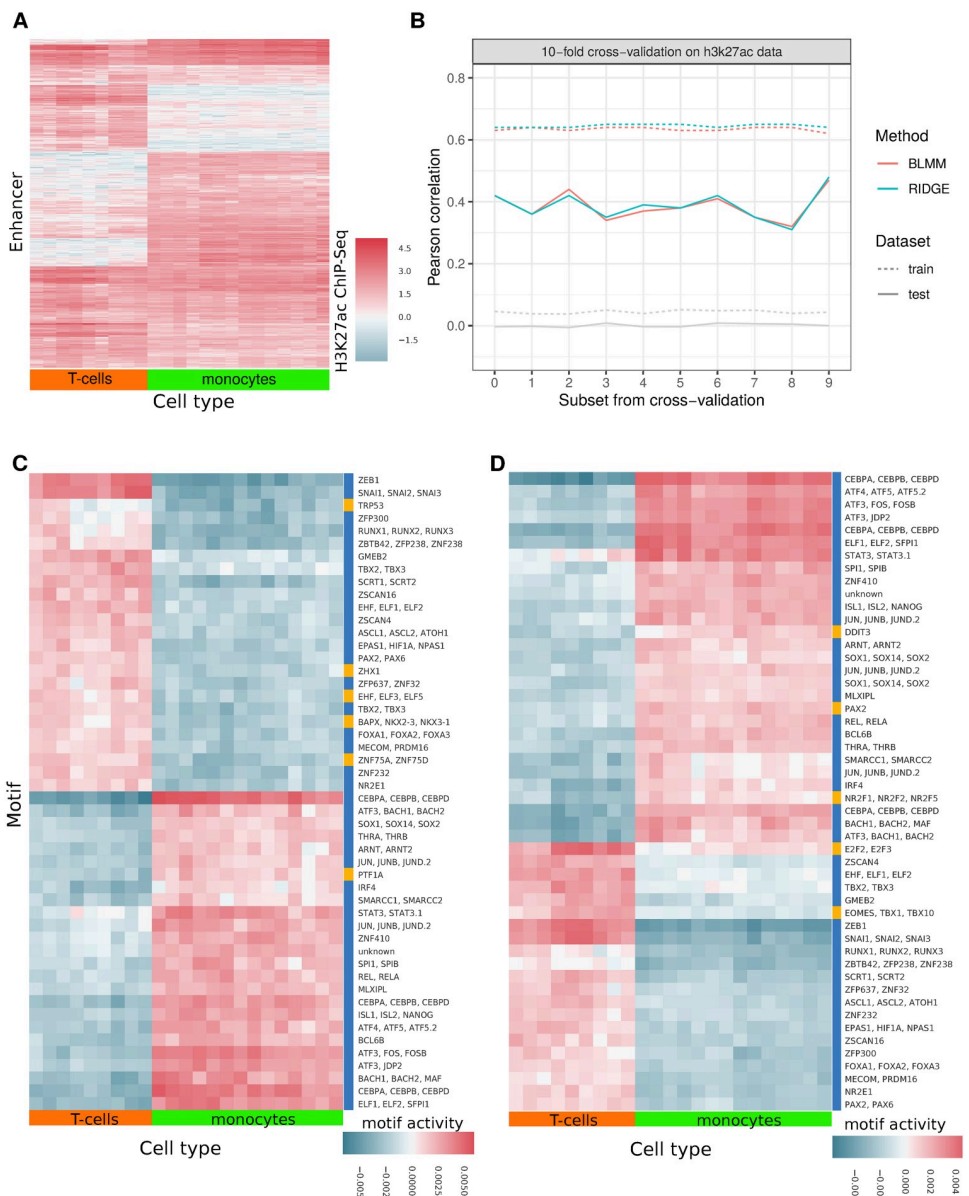

**Fig 3. Data application on H3K27ac data.** (A): Pattern of H3K27ac ChIP-seq signal in putative enhancers over two hematopoietic cell types, T cells and monocytes. (B): Pearson correlation between training set and predicted ChIP-seq signal on training set and between test set and predicted test ChIP-seq signal of a ten-fold cross-validation. In gray, the mean Pearson correlation between predicted and real signal using each method (overprinted lines show high concordance between methods), with a 1000-fold randomization of motif scores is shown. (C) and (D): Motif weights for 50 enhancers across the two cell types, assuming dependence (C) or independence (D) between the conditions. Enhancers are chosen based on largest difference in mean H3K27ac ChIP-seq signal per tissue group. Common motifs of Bayesian Linear Mixed Model (C) and Ridge Regression (D) are depicted in blue, others are depicted in yellow.

a strong correlation among the samples within the cell types, and independence, or even a slight anti-correlation, between the two cell types. This pattern follows the partitioning of the samples into biological replicates for the two different cell types.

Looking in more detail into the importance of the posterior motif influence $\hat{\omega}_{T,C}$, we compare the motifs by ranks. They are ordered based on the difference of the motif's mean

influence onto each class, i.e.

$$\bar{\omega}_{t,c} = \left| 1/C_{\text{monocytes}} \left( \sum_{\hat{c}=1}^{C_{\text{monocytes}}} \omega_{t,\hat{c}} \right) - 1/C_{\text{T-cells}} \left( \sum_{\tilde{c}=1}^{C_{\text{T-cells}}} \omega_{t,\tilde{c}} \right) \right|, \tag{6}$$

with $C_{\text{monocytes}} + C_{\text{T-cells}} = C$. They are sorted in decreasing order, i.e. the higher the mean difference, the higher the rank of a motif, where rank 1 is the most important motif and rank 623 the least important. For Fig 3C and 4D, we choose the top 50 ranking motifs, respectively. We color motifs that are shared for both methods in blue and different motifs in yellow. All in all, 40 motifs are shared among the 50 chosen. The posterior motif influences, depicted in Fig 3C and 4D, clearly separate the motif influences based on the underlying cell types for both models. Despite the clear differences in model estimates, the performances in predictive power are very similar. We hypothesize that the similarity of the performances results from the similarity between estimated covariance matrix $\mathbf{V}_C$ and noise $\Sigma_C$ in the Bayesian Linear Mixed Model, which leads to canceling out the covariance structure in the posterior. We elaborate on this phenomenon in the Discussion.

**RNA-seq data GTEx.** We further compare the performance of both model assumptions on a human tissue-specific RNA-seq dataset from GTEx. In comparison to the H3K27ac dataset discussed in the previous section, RNA-seq is known to exhibit more "technical" noise, i.e. noise that is added to the signal purely by conducting the experiment. Hence, the expected noise structure should be better separable from the motif signal.

For the $G = 5000$ most variable genes across the entire GTEx dataset, we apply the Bayesian Linear Mixed Model and the Ridge Regression on $C = 75$ randomly chosen samples. Among those, there are several biological replicates, resulting in 35 different tissue types across 21 organs. The results of the GTEx dataset are summarized in Fig 4.

Analogous to the analysis on the H3K27ac dataset, we conduct a ten-fold cross-validation together with a 1000 permutations over the motif scores $\mathbf{M}_{T,G}$ per cross-validation. The Pearson correlation values of the cross-validation of the expression data $\mathbf{Y}_{G,C}$ (colored lines) are very similar for both model assumptions (see Fig 4A) on the test (solid line) and on the training (dotted lines) dataset. Again, shuffling the motif scores along the genes negates the models' performances. The Pearson correlation of all motif scores between the two model assumptions per tissue is high with the median at 0.83, the first quantile at 0.79 and the third quantile at 0.90 (see Fig 4B). In Fig 4C we highlight examples of high (left) and low (remaining four panels on the right, same replicate) Pearson correlation values between the estimated motif weights from the two models (colored accordingly).

When we compare the estimated covariance matrices, there are strong differences in $\mathbf{V}_C$ and $\Sigma_C$ (S11–S14 Figs). Independent of the model assumptions, there is a difference of $10^4$ in order of magnitude of the signal assigned to the covariance between condition and the estimated noise.

We compute the inter-quantile range of the posterior motif influences of all $T = 623$ motifs per tissue to investigate the posterior motif influence $\hat{\omega}_{T,C}$ in more detail. We summarize the posterior motif influences over tissues with replicates with the median. We then filter those motifs that lie outside the range of 2.5 times the inter-quantile range above and below the median. Combining the two sets from Bayesian Linear Mixed Model and Ridge Regression results in 56 motifs, of which 50 are in the intersection of the two sets.

The clustering of posterior motif influence $\hat{\omega}_{T,C}$ results in comparable clusters (S10 Fig), with the clustering on tissues from Ridge Regression being seemingly better due to a clearer clustering of replicates.

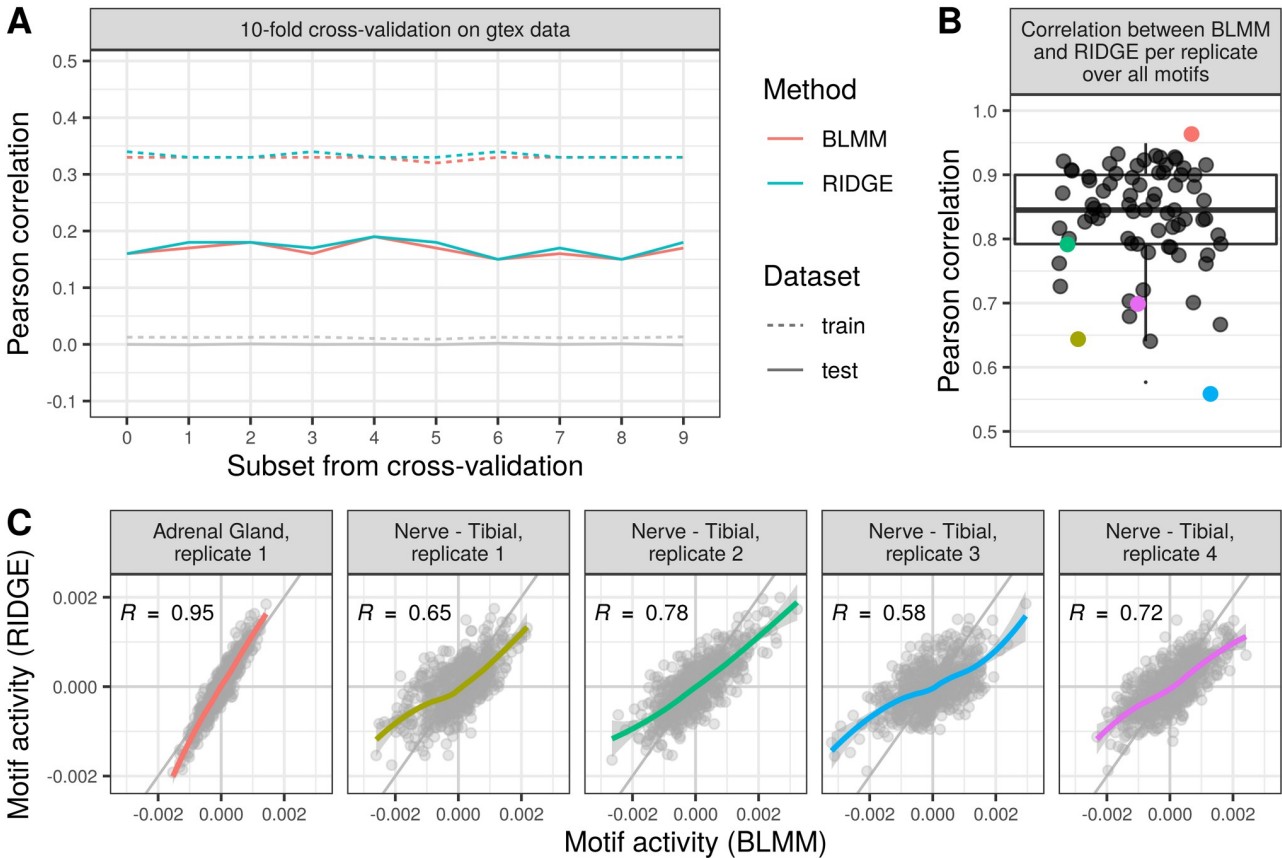

**Fig 4. Data application on GTEx data.** (A): Pearson correlation values between training set and predicted gene expression of training set and between the test set of the gene expression data and the predicted gene expression of ten-fold cross-validation for Bayesian Linear Mixed Model (BLMM, in red) and Ridge Regression (RIDGE, in blue). In gray, the mean Pearson correlation between predicted and real signal using both methods (overprinted lines show high concordance) with a 1000-fold randomization of motif scores is shown. (B): Pearson correlation of estimated motif scores on the basis of the Ridge Regression and the Bayesian Linear Mixed Model. Per correlation value, all motif scores are taken per replicate. Examples shown in (C) are colored according to respective color. (C): Exemplary scatter plots of tissue replicates, on which the Pearson correlation for the posterior motif influence $\hat{\omega}_{T,C}$ is high or low between the two methods, Bayesian Linear Mixed Model and Ridge Regression, as colored in (B). The values along the x-axis result from the Bayesian Linear Mixed Model, assuming dependence between samples, and on the y-axis from assuming independence (Ridge Regression). The diagonal is shown in gray and the overall trend of the data is shown with its 95% confidence interval.

The overall correlation between the posterior motif influence $\hat{\omega}_{T,C}$ per tissue of both methods is very high (median $\geq 0.95$, Fig 5A), which underlines the strong similarity between the two models (see S15 Fig for the five highest correlated motifs across all tissues and S16 Fig for the five lowest correlated motifs). The assumption of dependence between samples (BLMM) results in higher variation of posterior motif influences (see S17 Fig). Among the 56 chosen motifs, there are a few cases, where one model assumption yields a weight value of around zero and the other is significantly different from zero (see Fig 5B). Out of those four examples, we found evidence that the TFs are known to play a major role in the descriptive tissue: *FOXO1, FOXO3* and *FOXO4* in liver [48], *RUNX1-3* in blood [49–51] and in B-cell lymphocytes [52]. We found no explicit relevance for *YY1* in skin fibroblasts. The *YY1* TF is known to be involved in the repression and activation of a diverse number of promoters [53–55]. It is relatively highly expressed in skin cells and fibroblasts, which we find from gene expression profiles from the Protein Atlas [56], data available from v18.1 proteinatlas.org. This could be biologically relevant, but no specific function is known. Hence, despite similar performances we find evidence of motif influences found by the Bayesian Linear Mixed Model, but not by Ridge Regression. The same holds vice-versa for EBV-transformed lymphocytes, for which

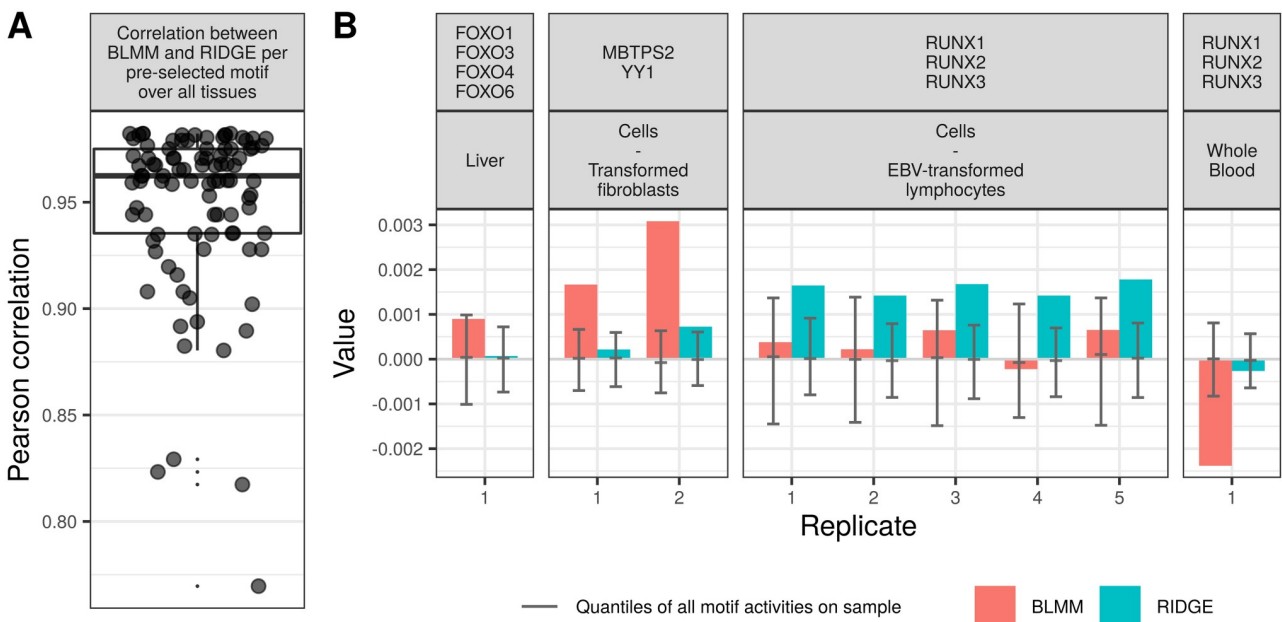

**Fig 5. IQR-study on GTEx motif-weight.** (A): Pearson correlation values between the predicted posterior motif influence $\hat{\omega}_{T,C}$ from the Bayesian Linear Mixed Model and the Ridge Regression of 56 selected motif values over all tissues. (B): extreme cases of different motif scores per method. Each box shows the predicted motif scores per sample, separated by model assumptions used to predict the scores (dependence, denoted BLMM, colored in red, and indepdence between samples, named RIDGE, colored in blue). Per sample, the first, second and third quantile (as in a boxplot) of the overall motif activity of all motifs on that sample are depicted in gray.

Ridge Regression predicts an influence of the motif to which the TFs *RUNX1, RUNX2, RUNX3* bind, whilst the Bayesian Linear Mixed Model does not. This finding is concordant with the observation that the Epstein-Barr virus (EBV) TF EBNA-2 induces RUNX3 expression in EBV-transformed lymphocytes [57].

**Time-series datasets.** As a final test of the Bayesian Linear Mixed Model we analyze two time-series datasets. Due to the nature of temporal expression data, we expect significant correlation of gene expression levels between subsequent time-points. We made use of two different time-series experiments: (i) reprogramming of fibroblasts to induced pluripotent stem cells (iPSCs) [41] and (ii) differentiation of keratinocytes [42] (see Methods for details). We analyze these datasets analogous to the previous benchmarks. We run a ten-fold cross-validation with 1000 permutations per fold to compare both model assumptions on each dataset. We summarize the most important results from the analysis in Fig 6, where we capture the Pearson correlation between expression data from the ten-fold cross-validation study (Fig 6A), as well as the Pearson correlation values between the two posterior motif influences $\hat{\omega}_{T,C}$ over all conditions (Fig 6B), which result from the two methods assuming (i) dependence as well as (ii) independence per time series.

We show the estimated covariance matrices $\mathbf{V}_C$ and $\Sigma_C$ for both model assumptions for the Cacchiarelli and Toufighi dataset in S19–S22 and S25–S28 Figs, respectively. Analogous to the previous two datasets, the covariance matrices from the Bayesian Linear Mixed Model show a correlation pattern of the data. Again, the values estimated for the noise matrix $\Sigma_C$ are larger by a factor of $10^4$ and more for both methods.

We depict scatter plots of the estimated motif weights $\hat{\omega}_{T,C}$ between the Bayesian Linear Mixed Model and Ridge Regression, separated by time frame, in S23 and S29 Figs.

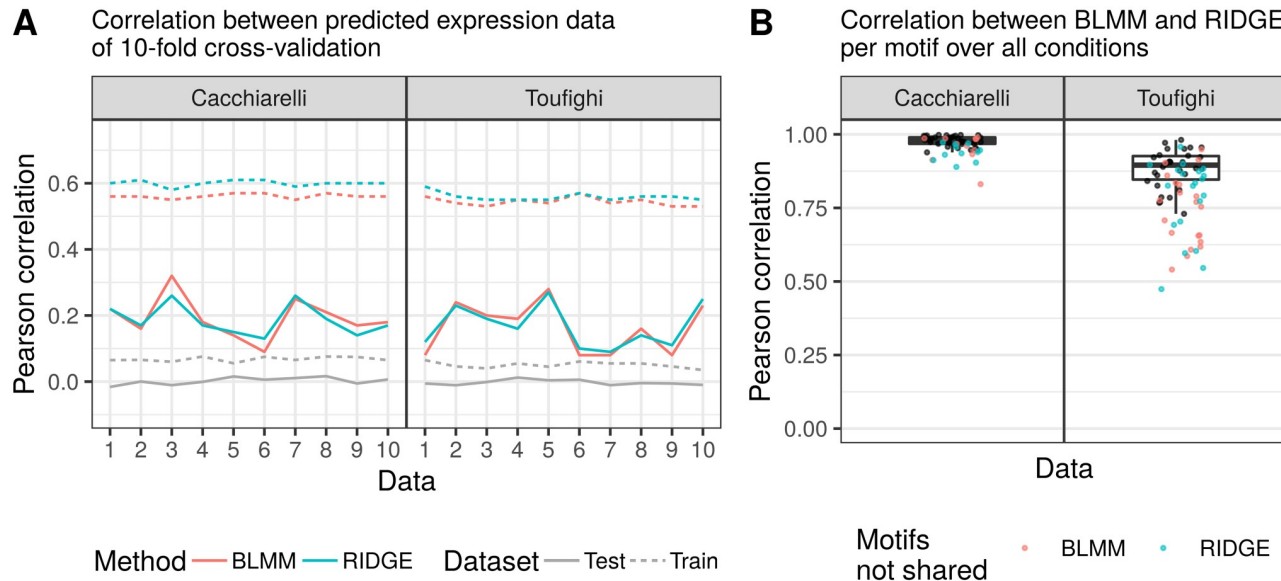

**Fig 6. Data application on time-series datasets.** (A): Pearson correlation between training set and predicted expression signal on training set and between test set and predicted test expression signal of a ten-fold cross-validation. In gray, the mean Pearson correlation between predicted and real signal using Bayesian Linear Mixed Model (results for Ridge Regression are almost identical) with a 1000-fold randomization of motif scores is shown. (B): Pearson correlation values per condition between the predicted motif weights $\hat{\omega}_{T,C}$ assuming (i) dependence and (ii) independence between the conditions.

Fig 6A shows the Pearson correlation between real and predicted expression levels for both the Cacchiarelli and Toufighi dataset. The correlation values average to 0.2 across the folds for the test dataset (ranging both from 0.1 to 0.3) and 0.6 for the training dataset. Both models show a similar degree of (over)fitting to the training data, and equal performance in predicting the test data. As expected, shuffling the motif scores (gray line) completely negates the model performance. Overall, we see that there is no significant difference between the two methods based on the cross-validation analysis. This recapitulates the findings from the previous benchmarks.

In Fig 6B, we depict the Pearson correlation values between the motif scores from both methods over all conditions (depicted in S18 and S24 Figs in detail). We only show the correlation values for the union of the 50 most variable motifs for both methods. The motifs that are in the common subset for both methods are colored in black. The remaining motifs are colored based on the method, for which they were found to be among the 50 most variable. There are 42 motifs shared among the 50 most variable motif scores across conditions for the Cacchiarelli dataset. The motif scores for the Cacchiarelli dataset are highly correlated with a median correlation value at 0.97 and the first and third quantile at 0.95 and 0.99, respectively. The correlation values rank from 0.83 to 0.998. Hence, both methods perform equally well on the Cacchiarelli dataset and produce very similar results. The motif scores of the Toufighi dataset are also highly correlated with a median correlation at 0.84, and the first and third quantile at 0.77 and 0.90, respectively. The estimated motif influences are more different between the two models for these data, as compared to the Cacchiarelli dataset. One of the reasons could be the nature of the data, as microarray data generally contains more technical noise and has a lower dynamic range as compared to RNA-seq data. While the predicted motif activity scores are different, qualitative evaluation of the identified motifs shows no clear advantage of either method (see S29 Fig). Motifs for TFs that are known to be differentially expressed in

keratinocyte differentiation show very similar score for both methods. For instance, TP63 is important for epidermal commitment and is downregulated during differentation, which is recapitulated by the predicted score of both methods. Similarly, both Ridge Regression and Bayesian Linear Mixed Model identify the JUN motif. The AP1 TFs that are known to bind to this motif are well-studied regulators of epidermal keratinocyte differentiation [58].

In summary, we can not find a clear advantage of Bayesian Linear Mixed Model as compared to Ridge Regression on the basis of these time-series datasets.

## Discussion

In this work we introduce a novel extension of Ridge Regression for motif activity analysis. The Bayesian Linear Mixed Model can leverage the sample covariation structure to more accurately determine motif activities. Through extensive simulation benchmarks, we observe a clear superiority of the Bayesian Linear Mixed Model over Ridge Regression when the entire signal can be explained by motif influence and no, or less, comparable structure between samples can be found in the noise. However, with the simulation study, we observe a decrease in performance for the Bayesian Linear Mixed Model for an increasingly structured signal in the noise (see S4 Fig). When applying the two model assumptions to four real-world datasets, we see no favorable performance on the basis of a cross-validation. Only a more detailed investigation of motif importance reveals some differences, which are a lot stronger for the GTEx and Cacchiarelli RNA-seq datasets, as more "technical" noise is present than in the acetylation dataset. The origin of noise in our model has two different sources: the "technical" noise, which includes noise introduced (i) in the lab (different technicians, different days of experiment conducted, pipeting error, different kits, etc.), (ii) by the machine (batch effect, sequencing, lane-to-lane variability, etc.), and (iii) through biological variability due to gene expression being a stochastic process [59]. The other source of noise originates from the model that explains the expression signal uniquely as a linear combination of motif scores. Other sources that contribute to the signal are not modeled here, hence end up in the noise. And it is this large contribution of noise to the signal that is no "technical" noise that causes the loss of performance of the dependence assumption in the covariance. Mathematically, $\mathbf{V}_C$ cancels out of the computation of the posterior (Eq (4)) if the covariance between conditions $\mathbf{V}_C$ and the noise term $\mathbf{\Sigma}_C$ are comparable. This is the reason why the Bayesian Linear Mixed Model does not perform better than the Ridge Regression in this case. If the noise takes a form similar to the correlation between conditions, say:

$$\mathbf{\Sigma}_C = \gamma \mathbf{V}_C, \tag{7}$$

with some constant scaling $\gamma$, then Eq (4) takes the following form:

$$\text{vec}(\hat{\omega}_{T,C}) = (\mathbf{V}_C \otimes \mathbf{M}_{T,G}^\mathsf{T} \mathbf{I}_T)[\sigma^2 \mathbf{V}_C \otimes \mathbf{\Pi}_G + \delta\gamma \mathbf{V}_C \otimes \mathbf{I}_G]^{-1} \text{vec}(\mathbf{Y}_{G,C}), \tag{8}$$

where $\mathbf{V}_C$ cancels out in the equation:

$$\text{vec}(\hat{\omega}_{T,C}) = (\mathbf{I}_C \otimes \mathbf{M}_{T,G}^\mathsf{T} \mathbf{I}_T)[\sigma^2 \mathbf{I}_C \otimes \mathbf{\Pi}_G + \delta\gamma \mathbf{I}_C \otimes \mathbf{I}_G]^{-1} \text{vec}(\mathbf{Y}_{G,C}). \tag{9}$$

Hence, the correlation structure between samples plays no role in the determination of posterior motif influence $\hat{\omega}_{T,C}$. One can therefore conclude, that the entire formulation of the Bayesian Linear Mixed Model we proposed is equivalent to Ridge Regression if $\mathbf{V}_C$ and $\mathbf{\Sigma}_C$ are the same up to a scaling factor. Hence, the Bayesian Linear Mixed Model is only to favor over Ridge Regression if there is less noise from the signal than from the "technical" noise in the data.

## Conclusion

In this research paper we extended a known framework to model motif influence on gene expression signal, which was originally introduced by [5, 6]. While the previous formulation assumes independence between samples, our Bayesian formulation provides the possibility to relax this assumption and allows to model correlation between samples and hence to better control the breakdown of measured signal into different sources. There are many applications in the field of molecular biology, where Ridge Regression has proved to be successful. The `limix` package itself was mainly developed to investigate the influence of SNPs on phenotype prediction [19, 60, 61]. With an increase of computational power and better implementations that reduce the computational complexity, this Bayesian formulation allows for a more flexible separation of source influences onto the signal.

We first ran a simulation study on the basis of which we showed a significant improvement of the Bayesian Linear Mixed Model as compared to Ridge Regression for data with independent noise. For noise that is dependent on the signal, the Bayesian Linear Mixed Model quickly loses its predictive power and has a similar performance to Ridge Regression. We further compared the two model assumptions on four real-world datasets: H3K27ac, RNA-seq and microarray data. Across all four datasets, we observed the same phenomenon as in the simulation study: no distinct superiority of the Bayesian Linear Mixed Model over Ridge Regression.

Practically, our findings indicate that the theoretical superior performance of the Bayesian Linear Mixed Model do not translate to noticeable improvements on motif activity estimation on real-world data. For expression data, we confirm the findings of earlier work, which demonstrated that at most 10-20% of gene expression levels can be explained by TF motifs near the gene promoter. Crucially, the remaining 80-90% of the variation is not independent noise. This expression variation contains, for instance, the regulatory effect of distal enhancers, RNA degradation rates and many other biological parameters that are not captured by our relatively simple model. This results in a similar covariance structure over samples in the expression modeled by the motifs and the signal that ends up in the noise term of the model. As we explain mathematically above, this means that the effect of the correlation structure will be canceled out. Even in the case of H3K27ac ChIP-seq data, where $\sim 40\%$ of the signal can be explained by TF motifs, we do not see a clear benefit of the Bayesian Linear Mixed Model over Ridge Regression.

In conclusion, with the current model formulation we observe that the Bayesian Linear Mixed Model does not gain predictive power over Ridge Regression using real-world data. However, this might potentially change if the formulation of the model's covariates is further improved. For instance, this could include incorporation of motifs at enhancer regions, chromatin interaction maps determined by chromosome conformation capture techniques such as Hi-C [62] and ChIP-seq assays measuring the chromatin environment.

Finally, while we showed here one specific application, we believe that these types of models can be more generally useful to model biological systems. The advancements made in faster implementations together with mathematical reformulations, as done by [18, 19], allow for the usage of more complex models, such as the Bayesian Linear Mixed Model over simple Ridge Regression. In concert with the increase in computational power, such increase of mathematical complexity becomes more feasible to work with and no longer represents a practical constraint as it used to.

## Supporting information

**S1 Fig. Ridge Regression is a special case of Bayesian Linear Mixed Model, limiting the estimated covariance and noise to be independent.** Data was generated over $G = 978$ informative

genes and $T = 623$ motif scores and 100 repetitions, with structured $\Sigma_C$. The results for Bayesian Linear Mixed Model and Ridge Regression (RIDGE, in blue) are equal when Bayesian Linear Mixed Model is limited to $\mathbf{V}_C = \sigma^2 \mathbf{I}_C$ and $\Sigma_C = \delta \mathbf{I}_C$ (BLMM_id, in red). The Pearson correlation values are computed between the generated and predicted posterior motif influence $\hat{\omega}_{T,C}$, and separated by method used to compute them. Data is generated with (i) independent samples, $\mathbf{V}_C = \mathbf{I}_C$, (ii) unrestricted correlation between samples, (iii) 50% correlated data, where the samples cluster in many ($k = \frac{1}{2} C$) sample groups, (iv) highly correlated data, by generating a covariance matrix with $k = 2$ completely correlated sample groups, with $C$ the number of conditions.
(PDF)

**S2 Fig. Simulation study with Spearman's rank correlation values.** Results of simulation study analogously presented as in Simulation with Spearman's rank correlation values.
(PDF)

**S3 Fig. Simulation study with Mean-Squared Error values.** Results of simulation study analoguously presented as in Simulation with Mean-Squared Error values.
(PDF)

**S4 Fig. Models' performance interchange superiority for data generated with increasing structuredness in noise.** Data was generated over $G = 978$ informative genes, $T = 623$ motif scores, 100 repetitions, and with a covariance matrix "$\mathbf{V}_C$: highly correlated (two groups)". $\Sigma_C$ is generated with increasing degree of structuredness (x-axis). Model performance of Bayesian Linear Mixed Model (BLMM, in red) and Ridge Regression (RIDGE, in blue) are shown on the bases of the Pearson correlation values between generated and predicted motif-condition-weights.
(PDF)

**S5 Fig. Simulation study on $G = 5000$ genes.** We compare the model assumptions of Bayesian Linear Mixed Model (BLMM, depicted in red) and Ridge Regression (LIMIX, in blue) on datasets generated with $C = 50$ samples, and $G = 5000$ genes. On the x-axis we show data that is generated with unstructured noise (degree of structuredness = 0) and with structured noise with $\rho = 0.7$. On the y-axis, we depict the Pearson correlation values between generated and predicted motif-condition weights. In each panel, the data was generated with different assumptions on the degree of correlation between samples: (i) independence ($\mathbf{V}_C = \mathbf{I}_C$ (upper left), (ii) unrestricted correlation (upper right), (iii) correlated with many sample groupgs (lower left), and (iv) highly correlated with two sample groups) (lower right). There is no difference in performance when increasing the dimensionality of genes. The Bayesian Linear Mixed Model has predictive power over Ridge Regression when the data is correlated, uniquely for unstructured noise. For structured noise ($\rho = 0.7$), there is no gain in performance, despite the bigger size of the dataset.
(PDF)

**S6 Fig. H3K27ac: $\mathbf{V}_C$ for Ridge regression.** Estimated correlation between conditions $\mathbf{V}_C$ assuming independence between the conditions for the H3K27ac dataset.
(PDF)

**S7 Fig. H3K27ac: $\Sigma_C$ for Ridge regression.** Estimated noise $\Sigma_C$ assuming independence between the conditions for the H3K27ac dataset.
(PDF)

**S8 Fig. H3K27ac: $\mathbf{V}_C$ for Bayesian Linear Mixed Model.** Estimated correlation between conditions $\mathbf{V}_C$ assuming dependence between the conditions for the H3K27ac dataset.
(PDF)

**S9 Fig. H3K27ac: $\Sigma_C$ for Bayesian Linear Mixed Model.** Estimated noise $\Sigma_C$ assuming dependence between the conditions for the H3K27ac dataset.
(PDF)

**S10 Fig. GTEx: Clustermaps of motif-condition-weight matrix.** Weights for the motif-condition-weights of those 56 common motifs, that are outside of 2.5 times the inter-quantile range for all motifs over a tissue. The weights are computed assuming dependence (A) or independence (B) between the conditions.
(PDF)

**S11 Fig. GTEx: $\mathbf{V}_C$ for Ridge regression.** Estimated correlation between conditions $\mathbf{V}_C$ assuming independence between the conditions for the GTEx dataset.
(PDF)

**S12 Fig. H3K27ac: $\Sigma_C$ for Ridge regression.** Estimated noise $\Sigma_C$ assuming independence between the conditions for the GTEx dataset.
(PDF)

**S13 Fig. GTEx: $\mathbf{V}_C$ for Bayesian Linear Mixed Model.** Estimated correlation between conditions $\mathbf{V}_C$ assuming dependence between the conditions for the GTEx dataset. Note that only a subset of samples are labeled.
(PDF)

**S14 Fig. GTEx: $\Sigma_C$ for Bayesian Linear Mixed Model.** Estimated noise $\Sigma_C$ assuming dependence between the conditions for the GTEx dataset. Note that only a subset of samples are labeled.
(PDF)

**S15 Fig. GTEx: Most similar motif scores between methods.** Motif values for the five highest correlated motif scores over all tissues. Note that only a subset of samples are labeled.
(PDF)

**S16 Fig. GTEx: Least similar motif scores between methods.** Motif values for the five lowest correlated motif scores over all tissues.
(PDF)

**S17 Fig. GTEx: High variation between replicates.** Variation of all chosen 56 motif scores on exemplary tissue EBV—cellline. Note that only a subset of samples are labeled.
(PDF)

**S18 Fig. Cacchiarelli: Clustermaps of motif-condition-weight matrix.** Motif weights for the 50 most variable weights across conditions assuming dependence (A) or independence (B) between the conditions. Common motifs of Bayesian Linear Mixed Model (A) and Ridge Regression (B) are depicted in blue, others are depicted in yellow.
(PDF)

**S19 Fig. Cacchiarelli: $\mathbf{V}_C$ for Ridge Regression.** Estimated correlation between conditions $\mathbf{V}_C$ assuming independence between the conditions for the Cacchiarelli dataset.
(PDF)

**S20 Fig. Cacchiarelli: $\Sigma_C$ for Ridge Regression.** Estimated noise $\Sigma_C$ assuming independence between the conditions for the Cacchiarelli dataset.
(PDF)

**S21 Fig. Cacchiarelli: $\mathbf{V}_C$ for Bayesian Linear Mixed Model.** Estimated correlation between conditions $\mathbf{V}_C$ assuming dependence between the conditions for the Cacchiarelli dataset.
(PDF)

**S22 Fig. Cacchiarelli: $\Sigma_C$ for Bayesian Linear Mixed Model.** Estimated noise $\Sigma_C$ assuming dependence between the conditions for the Cacchiarelli dataset.
(PDF)

**S23 Fig. Cacchiarelli: Scatterplot of estimated motif weights $\boldsymbol{\omega}_{T,C}$.** Scatterplot of posterior motif weights $\boldsymbol{\omega}_{T,C}$ of Bayesian Linear Mixed Model vs. Ridge Regression, depicted per time series.
(PDF)

**S24 Fig. Toufighi: Clustermaps of motif-condition-weight matrix.** Motif weights for the 50 most variable weights across conditions assuming dependence (A) or independence (B) between the conditions. Common motifs of Bayesian Linear Mixed Model (A) and Ridge Regression (B) are depicted in blue, others are depicted in yellow.
(PDF)

**S25 Fig. Toufighi: $\mathbf{V}_C$ for Ridge Regression.** Estimated correlation between conditions $\mathbf{V}_C$ assuming independence between the conditions for the Toufighi dataset.
(PDF)

**S26 Fig. Toufighi: $\Sigma_C$ for Ridge Regression.** Estimated noise $\Sigma_C$ assuming independence between the conditions for the Toufighi dataset.
(PDF)

**S27 Fig. Toufighi: $\mathbf{V}_C$ for Bayesian Linear Mixed Model.** Estimated correlation between conditions $\mathbf{V}_C$ assuming dependence between the conditions for the Toufighi dataset.
(PDF)

**S28 Fig. Toufighi: $\Sigma_C$ for Bayesian Linear Mixed Model.** Estimated noise $\Sigma_C$ assuming dependence between the conditions for the Toufighi dataset.
(PDF)

**S29 Fig. Toufighi: Scatterplot of estimated motif weights $\boldsymbol{\omega}_{T,C}$.** Scatterplot of posterior motif weights $\boldsymbol{\omega}_{T,C}$ of Bayesian Linear Mixed Model vs. Ridge Regression, depicted per time series.
(PDF)

**S1 Appendix. A. Bayesian Linear Mixed Models.** We give a more detailed derivation of the model introduced in Eqs (1)–(4), providing more information about the Bayesian formulation.
(PDF)

**S2 Appendix. B. Simulation.** Extensive details about the simulation of data.
(PDF)

## Acknowledgments

Parts of this work were carried out on the Dutch national e-infrastructure with the support of SURF Cooperative.

## Author Contributions

**Conceptualization:** Cornelis A. Albers.

**Formal analysis:** Simone Lederer.

**Methodology:** Simone Lederer, Simon J. van Heeringen, Cornelis A. Albers.

**Software:** Simone Lederer.

**Supervision:** Tom Heskes, Simon J. van Heeringen, Cornelis A. Albers.

**Writing – original draft:** Simone Lederer.

**Writing – review & editing:** Simone Lederer, Tom Heskes, Simon J. van Heeringen, Cornelis A. Albers.

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
