## [Decision Letter · Decision Letter 0]

25 Nov 2019

PONE-D-19-28028

Bayesian Linear Mixed Models for Motif Activity Analysis

PLOS ONE

Dear Mrs Lederer,

Thank you for submitting your manuscript to PLOS ONE. After careful consideration, we feel that it has merit but does not fully meet PLOS ONE’s publication criteria as it currently stands. Therefore, we invite you to submit a revised version of the manuscript that addresses the points raised during the review process.

Both reviewers appreciated the question of modeling correlation between samples in predicting expression from motif activity, however, there are several points that needs to be addressed in the revision to improve the technical soundness of the paper. Please address the comments raised by both reviewers, and importantly address the following points.

1. Please clarify the novel versus existing work in this manuscript.

2. Please consider addition of experiments on an additional dataset, namely, a time series dataset (as suggested by Reviewer 2).

3. Please provide details about the selected parameter settings.

We would appreciate receiving your revised manuscript by Jan 09 2020 11:59PM. To enhance the reproducibility of your results, we recommend that if applicable you deposit your laboratory protocols in protocols.io, where a protocol can be assigned its own identifier (DOI) such that it can be cited independently in the future. For instructions see: http://journals.plos.org/plosone/s/submission-guidelines#loc-laboratory-protocols

We look forward to receiving your revised manuscript.

Kind regards,

Sushmita Roy, Ph.D.

Academic Editor

PLOS ONE

Journal Requirements:

Reviewers' comments:

Reviewer's Responses to Questions

**Comments to the Author**

1. Is the manuscript technically sound, and do the data support the conclusions?

Reviewer #1: Partly

Reviewer #2: Partly

2. Has the statistical analysis been performed appropriately and rigorously? 

Reviewer #1: Yes

Reviewer #2: Yes

3. Have the authors made all data underlying the findings in their manuscript fully available?

Reviewer #1: Yes

Reviewer #2: Yes

4. Is the manuscript presented in an intelligible fashion and written in standard English?

Reviewer #1: Yes

Reviewer #2: Yes

5. Review Comments to the Author

Reviewer #1: The paper notes that most models relating TF motif activity to gene expression

level assume that samples are independent of each other -- an assumption that

may not be satisfied for real datasets. They take a linear model of TF

activity-gene expression prediction and incorporate a prior on the TF motif

activity. They show that when the expression noise between samples is

unstructured, modeling the correlation between samples yields better results.

However, when the noise has a similar structure as the correlation between

samples, there's little to no advantage in modeling the correlation.

The question of whether modeling the correlation between samples is necessary

is an important one. The approach the authors take of directly testing the

performance of a particular model with and without the correlation modeled is

sound. They prove that if the noise correlation between samples is equal to the

total correlation upto a scaling factor, their model is provably equivalent to

Ridge Regression. The authors find that modeling correlation on two real world

datasets is not important.

Major concerns:

(1) The title of the manuscript seems to indicate they are introducing a novel

useful model for motif analysis. However the novelty is limited and the model

is not useful on real data as they state in the abstract.

(2) The number of real datasets tested on is only 2 which makes any implication

that modeling correlation is not useful suspect.

(3) The authors could test their method on time series data. There are several

models that successfully incorporate temporal dependence and it would be a good

test of their methodology to check if they find no advantage in modeling the

correlation there as well.

(4) The story needs to be made more clear. The manuscript should either be geared

towards the introduction of a new model and showing its advantages or testing whether

modeling the dependence between samples helps in which case it would be useful to test

more models with and without the dependence modeling as well as test on more datasets.

Testing on time series data may be a good control.

Reviewer #2: The authors compared Bayesian ridge regression and Bayesian Linear Mixed Model based motif activity analysis (modeling influence of transcription factors on gene expression on the basis of TF motifs in cis-regulatory regions). More specifically, they investigate whether a Bayesian model that allows for correlations results in more accurate inference of motif activities. They demonstrated that there is no advantage to using the Bayesian Linear Mixed Model if the noise has a similar covariance structure over samples.

It would be helpful if the authors explained to the reader what is novel and what is confirmatory.

Specific comments

1. Results:

a. Authors didn’t elaborate on how they chose parameters for models.

b. How robust is results for both models?

c. Can authors randomize motif hits on real datasets and compare performance for both methods?

d. Can authors report spearman correlation as well as MSE performance?

e. Result section would be benefit from the addition of another dataset where the ground truth is know and compare methods

2. Figures:

a. Please give a description about the framework in Figure 1 legend.

6. PLOS authors have the option to publish the peer review history of their article (what does this mean?). If published, this will include your full peer review and any attached files.

Reviewer #1: No

Reviewer #2: No

---

## [Author Response · Author response to Decision Letter 0]

3 Mar 2020

Dear Reviewers,

We would like to submit our revised manuscript entitled Investigating the Effect of Dependence between Conditions in Bayesian Linear Mixed Models for Motif Activity Analysis by Simone Lederer, Simon J van Heeringen, Tom Heskes and Cornelis A Albers.

We thank the reviewers for their comments which we believed helped to make the paper more understandable, highlight the results better and helps the reader to navigate better through the paper.

We have carefully taken all comments into consideration and would like to address them in this letter and explain to you in detail how we proceeded.

We submit two versions of the revised article. The first is the revised version of the article. In the second we highlight all changes made in the text. Please note that we did not highlight changes in tables or figures. We hope that this second version helps the reviewers better to identify the changes we made based on their remarks.

Reviewer 1

Major concerns

 1. The title of the manuscript seems to indicate they are introducing a novel useful model for motif analysis. However the novelty is limited and the model is not useful on real data as they state in the abstract.

We agree with the reviewer, that the original title “Bayesian Linear Mixed Models for Motif Activity Analysis” might have been misleading. We changed the title to “Investigating the Effect of Dependence between Conditions in Bayesian Linear Mixed Models for Motif Activity Analysis”.

 2. The number of real datasets tested on is only 2 which makes any implication that modeling correlation is not useful suspect.

We introduced two more datasets (details below in the following paragraph) and show that the results from the simulation study do indeed hold true on real datasets. We would also like to point out that we have a mathematical explanation for why modeling correlation doesn’t work well in practice. 

 3. The authors could test their method on time series data. There are several models that successfully incorporate temporal dependence and it would be a good test of their methodology to check if they find no advantage in modeling the correlation there as well.

We thank the reviewer for the suggestion. We additionally analyzed two datasets on time series data. We chose two datasets which measure gene expression changes of in vitro cultured cells of two well-characterized systems, RNA-seq data of transdifferentiation from fibroblast to pluripotent cells [Cacchiarelli et al, 2015] and differentiation of keratinocytes measured using micro-array data [Toufighi et al., 2015]. 

 4. The story needs to be made more clear. The manuscript should either be geared towards the introduction of a new model and showing its advantages or testing whether

modeling the dependence between samples helps in which case it would be useful to test more models with and without the dependence modeling as well as test on more datasets.

Testing on time series data may be a good control.

In our proposed article we focus on the extension of a widely used model, the Ridge Regression, to allow for dependencies between conditions. We think it is important to introduce the model extensively for the reader to be able follow up and understand the results, that are presented. We do extensively test the model on four datasets in total and show that the results of the simulation study are consistent with real world data. We additionally give an explanation to why the Bayesian Linear Mixed Model does not perform significantly better. We consider this detail to be of importance for other researchers to consider in further modeling approaches.

Reviewer 2

Major comments:

 1. It would be helpful if the authors explained to the reader what is novel and what is confirmatory.

We developed the mathematical framework to allow for correlation/covariance between samples and applied it to predicting motif activity. Predicting motif activity has hitherto only been done with Ridge Regression. We rewrote parts of the discussion and conclusion to better reflect this.

Specific comments

 2. Results:

 a. Authors didn’t elaborate on how they chose parameters for models.

We stated the parameter settings chosen in section Implementation and model testing. For a better navigation, we separated this section into three different sections, Code availability, Model fitting, and Visualization. In Model fitting, we give, to the best of our knowledge, all important details about the initial parameter settings and their computation.

 b. How robust is results for both models?

We used 100 repetitions in the simulation study, where the ground truth is known, and apply cross-validation on both datasets where the ground truth is not known, to show the robustness of both models.

 c. Can authors randomize motif hits on real datasets and compare performance for both methods?

We now show results for randomized motif scores in every analysis. As expected, randomized motifs have no predictive performance, as measured in the test data set during cross-validation.

 d. Can authors report spearman correlation as well as MSE performance?

We included the Spearman rank correlation and MSE values in the Supplement, Fig. S8, Fig, S9 

 e. Result section would be benefit from the addition of another dataset where the ground truth is know and compare methods

see comments 2-4 of reviewer 1. It is not possible to find datasets with known ground-truth, but we extended the analysis by two more datasets that incorporate time series. The lack of ground truth data for gene regulatory network analysis is a problem in the field. While we wholeheartedly agree with the reviewer that this would be highly beneficial, it falls outside the scope of this work to generate such a data set.

 3. Figures:

 a. Please give a description about the framework in Figure 1 legend.

We thank the reviewer for the comment and included a description to the project overview in Figure 1.

We hereby hope to have clarified all concerns and would like to thank the reviewers for their comments, which we believe supported the credibility of the results, made the paper more understandable, and will help the reader to better navigate through the text.

Best regards,

Simone Lederer

---

## [Decision Letter · Decision Letter 1]

2 Apr 2020

Investigating the Effect of Dependence between Conditions with Bayesian Linear Mixed Models for Motif Activity Analysis

PONE-D-19-28028R1

Dear Dr. Lederer,

We are pleased to inform you that your manuscript has been judged scientifically suitable for publication and will be formally accepted for publication once it complies with all outstanding technical requirements.

With kind regards,

Sushmita Roy, Ph.D.

Academic Editor

PLOS ONE

Additional Editor Comments (optional):

Reviewers' comments:

Reviewer's Responses to Questions

**Comments to the Author**

1. If the authors have adequately addressed your comments raised in a previous round of review and you feel that this manuscript is now acceptable for publication, you may indicate that here to bypass the “Comments to the Author” section, enter your conflict of interest statement in the “Confidential to Editor” section, and submit your "Accept" recommendation.

Reviewer #1: All comments have been addressed

Reviewer #2: All comments have been addressed

2. Is the manuscript technically sound, and do the data support the conclusions?

Reviewer #1: Partly

Reviewer #2: Yes

3. Has the statistical analysis been performed appropriately and rigorously? 

Reviewer #1: Yes

Reviewer #2: Yes

4. Have the authors made all data underlying the findings in their manuscript fully available?

Reviewer #1: Yes

Reviewer #2: Yes

5. Is the manuscript presented in an intelligible fashion and written in standard English?

Reviewer #1: Yes

Reviewer #2: Yes

6. Review Comments to the Author

Reviewer #1: I appreciate the authors addressing all my comments and being honest about the results. While their results on the time series data are not encouraging, their analysis of their method is thorough and it may be a useful analysis to have available to future researchers.

Reviewer #2: Authors improved their analysis and manuscript text. They addressed my concerns. I don't have any additional comments.

7. PLOS authors have the option to publish the peer review history of their article (what does this mean?). If published, this will include your full peer review and any attached files.

Reviewer #1: No

Reviewer #2: No

---

## [Editor Report · Acceptance letter]

7 Apr 2020

PONE-D-19-28028R1 

Investigating the Effect of Dependence between Conditions with Bayesian Linear Mixed Models for Motif Activity Analysis 

Dear Dr. Lederer:

I am pleased to inform you that your manuscript has been deemed suitable for publication in PLOS ONE. Congratulations! Your manuscript is now with our production department. 

With kind regards,

on behalf of

Dr. Sushmita Roy 

Academic Editor

PLOS ONE